# A Proposal for Formulating a Spectrum Usage Fee for 5G Private Networks in Indonesian Industrial Areas

**Alfin Hikmaturokhman** [1] , **Kalamullah Ramli** [1,*] , **Muhammad Suryanegara** [1] , **Anak Agung Putri Ratna** [1] , **Ibrahim Kholilul Rohman** [2] **and Moinul Zaber** [3]

1 Department of Electrical Engineering, Universitas Indonesia, Depok 16424, Indonesia; alfin.hikmaturokhman@ui.ac.id (A.H.); m.suryanegara@ui.ac.id (M.S.); ratna@eng.ui.ac.id (A.A.P.R.)
2 School of Strategic and Global Studies, Universitas Indonesia, Jakarta Pusat 10430, Indonesia; Ibrahim.rohman@gmail.com
3 UNU-EGOV, United Nations University Operating Unit on Policy-Driven Electronic Governance, 4810-445 Guimarães, Portugal; zaber@unu.edu
* Correspondence: kalamullah.ramli@ui.ac.id

**Abstract:** The Indonesian spectrum usage fees—the so-called Biaya Hak Pengguna Frekuensi Izin Pita Frekuensi Radio (BHP IPFR)—are currently calculated using a formula determined by the three following main parameters: the frequency band, the country's economic parameter, and the nationwide population. As spectrum usage fees are proportional to the width of the bandwidth, the current formula would result in an extremely high price when applied to 5G-mmWave private networks, with the cost burden being a direct consequence for the service operator. In this paper, we propose the formulation of a new spectrum usage fee for 5G-mmWave private network implementation in Indonesian industrial areas. To do so, we evaluate the current formula, adopt the framework offered by the ITU-R SM.2012-5 (06/2016), and use an industrial reference index—the Indonesia Industry Readiness Index 4.0 (INDI 4.0) score. We test the proposal by applying the new formula to calculate the 5G-mmWave private network spectrum usage fee for the Jakarta industrial area. The result shows that the new formula gives a lower spectrum usage fee than the current formula, which benefits 5G-mmWave private network service operators. Such savings can be regarded as a government subsidy for the service operators to use in various ways in the industry, providing further economic benefits. Using the input–output model, we prove that despite the proposed new formula brings a lower spectrum usage fee, resulting in a loss in state income, it will lead to a much greater positive impact on the national economic output. Applying the new formula will eventually have a multiplier effect on various sectors and encourage digital economic growth and national digital transformation, especially for vertical industries in Indonesia. This study may serve as a guideline or initial reference for Indonesian policymakers and service operators for applying the CAPEX and OPEX cost of using the new spectrum for 5G-mmWave private network service implementation and estimating the economic multiplier for 5G-mmWave private network service deployment in industrial areas. It can also be used as a benchmark case for other countries to apply spectrum usage fees for private networks in industrial areas.

**Keywords:** 5G-mmWave; spectrum usage fee; 5G private network; input–output model; economic impact of 5G

## 1. Introduction

Spectrum issues are said to be one of the most important parameters in 5G deployment [1], ranging from on which frequency 5G should be allocated to how much of the spectrum usage fee should be paid by 5G service operators. Spectrum allocation affects network capacity and coverage, directly impacting the service deployment cost and service operators' potential income [2]. At the same time, the spectrum is considered state income, with service operators paying spectrum usage fees to the government.

In Indonesia, the spectrum policy is regulated by the Ministry of Communication and Information (Kementrian Komunikasi dan Informatika), by which the government obliges any service operator to pay the spectrum usage fee (Biaya Hak Pengguna Frekuensi Izin Pita Frekuensi Radio [BHP IPFR]) [2]. The Ministry has a great interest in generating more state income from the telecommunications sector, which accounted for 5.17% of the total Indonesian gross domestic product (GDP) in 2019 [3].

The current formula for calculating Indonesian spectrum usage fees is determined by the three following main parameters: the frequency band, the country's economic parameters, and the nationwide population. The formula has been applied since 2010, primarily to calculate fees for the usage of 2G, 3G, and 4G network spectrums. As 2G, 3G, and 4G are intended to only provide data communication services, the current formula simply takes into account the population size without considering the area, whether in residential, commercial, or industrial zones.

Previous studies on spectrum usage fees may be found in [4–6], which look at how such fees are calculated for service operators (2G/3G) in Taiwan and Indonesia. These countries' formulas differ based on various factors, including rivalry among service providers, geographic location, population, bandwidth, and expected government income from spectrum usage fees, among others. However, these studies have not discussed a spectrum usage fee that uses technology to access an mmWave frequency band with extensive bandwidth and a limited distance.

In other studies [7–10], the authors only compare different pricing methods for private LTE and 5G networks using Finland as an example country, but with a narrowband bandwidth (10 MHz) and mid-band frequency (3.5 GHz). The present study only discusses spectrum usage fees for technologies that use mid-band frequencies (3.5 GHz) with limited bandwidth (10 MHz) and the national population. In contrast, 5G will be optimally used with a minimum bandwidth of around 100 MHz mmWave as a use case for industrial areas with a limited coverage area and limited population.

The current formula does not work well for calculating 5G-mmWave private network spectrum usage fees. Because the spectrum usage fee is proportional to the large bandwidth and uses the national population, the current formula would result in an excessive price when applied to 5G-mmWave private networks (26–28 GHz), which require a bandwidth of about 100 MHz. From a technical perspective, the deployment cost is increased as a transmitter station for 5G-mmWave private networks covers a very small area or has limited coverage and may only serve a specific population. In the end, such a condition may endanger the financial balance of the 5G-mmWave private network service operator.

However, there is a potential advantage to using 5G-mmWave as a private network in industrial areas. Private networks rely on 5G-mmWave to provide more limited coverage, providing an ideal technological platform for a factory's operation and production line. Service operators may utilize such an opportunity to create a 5G-mmWave private network, making it a new source of state income. This state income will also help increase the output of the national economy over the next few years [11].

For this reason, Indonesia needs a new formula for spectrum usage fees that would result in a more financially acceptable and affordable 5G-mmWave private network service operator. This paper proposes a new formula for calculating the spectrum usage fee for mmWave spectrum implementation in industrial areas. The main basis for developing the proposed new formula is by adopting the Indonesia Industry Readiness Index 4.0 (INDI 4.0) score for five industrial sectors prioritized in Indonesia. After considering this factor, we also take into account the number of workers in the industry and combine them with the ITU-R SM.2012-5 (06/2016) framework.

We test the proposal by applying the new formula to the Jakarta industrial area, assuming that 5G-mmWave would be deployed as a private network serving two industrial areas: Pulogadung and Kawasan Berikat Nusantara (KBN).

It is found that the proposed new formula is advantageous in that the fee would always be lower than that given by the current formula. Hence, the lower fee can be

regarded as a subsidy for the prospective 5G-mmWave private network service operators. However, because the government would expect state income, such a reduction should be compensated for by other benefits to the country. Accordingly, we also developed the input–output model (I–O) to estimate the economic impact of 5G-mmWave private networks at the national level as a contribution to the national Indonesian economic output.

This paper makes the following key contributions:

- It proposes the formulation of a new spectrum usage fee for 5G-mmWave private network implementation in Indonesian industrial areas.
- The framework can be used as a recommendation for Indonesian regulatory policy-makers, as the country will start deploying 5G-mmWave private networks in the near future. The proposed new formula reflects a comprehensive policy for supporting industrialization and digital transformation and enables us to estimate the economic multiplier for 5G-mmWave private network deployment in industrial areas.
- The new spectrum usage fee approach provides an easy and direct way to price spectra, so it can be used as a benchmark for other countries to apply spectrum usage fees to 5G-mmWave private networks in industrial area.

The remainder of the paper is organized as follows: Section 2 explains the underlying theories. Section 3 focuses on the Indonesian profile, including its regulatory industrial ecosystem. Section 4 evaluates the current formula for spectrum usage fees in Indonesia, while Section 5 constitutes the Methods and Proposed New Formula. Section 6 focuses on the Results. Section 7 is a discussion of the finds, and finally, Section 8 concludes the study and suggests future studies.

## 2. Underlying Theories

### 2.1. 5G-mmWave Technology

A clear phenomenon of increased data traffic shows that mobile cellular services have become essential for providing very rapid access to information in support of human activities and the increasing quality of life. Global mobile data traffic is projected to increase by a factor of close to 5 to 164EB per month by 2025 [12]. The deployment of previous mobile technologies (2G/3G/4G) is no longer able to meet human needs, so 5G, as a new technology, requires greater bandwidth availability.

Technically, the 5G frequency band is divided into FR 1, which works in the frequency range below 6 GHz, and FR 2, which works in the 24–52 GHz frequency range. The FR 2 frequency band is called 5G-mmWave and is in accordance with ITU-WRC-19. 5G development on mmWave will be operated in FR 2 frequency bands, including 26 GHz and 28 GHz.

As seen in Figure 1, according to the International Telecommunication Union's (ITU) ITU-R M.2083-0 recommendation, the potential 5G services can be grouped into the three following usage scenarios: enhanced mobile broadband, ultra-reliable, and low-latency and massive machine-type communications [13].

Focusing on the use case scenario of enhanced mobile broadband, 5G technology should be operated on a wider bandwidth to establish use cases [14]. With regard to the 5G standard release 15, the peak data rate (Mbps) can be calculated on the basis of (1) [15], with the peak data rate showing a directly proportional relationship with the bandwidth.

$$\text{Data rate (Mbps)} = 10^{-6} \times \sum_{j=1}^{J} \left( v_{Layers}^{(j)} \times Q_m^{(j)} \times f^{(j)} \times R_{max} \times \frac{N_{PRB}^{BW(j),\,\mu} x12}{T_s^\mu} \times \left( 1 - OH^{(j)} \right) \right) \tag{1}$$

Equation (1) shows that the 5G peak data rate formula is calculated using the factors of bandwidth ($BW$), the number of aggregated component carriers in a band or band combination ($j$), the maximum number of layers ($v_{Layers}^{(j)}$), the type of maximum modulation order ($Q_m$), and the numerology of carrier spacing ($\mu$) used in the Orthogonal Frequency Division Multiplexing (OFDM) technique.

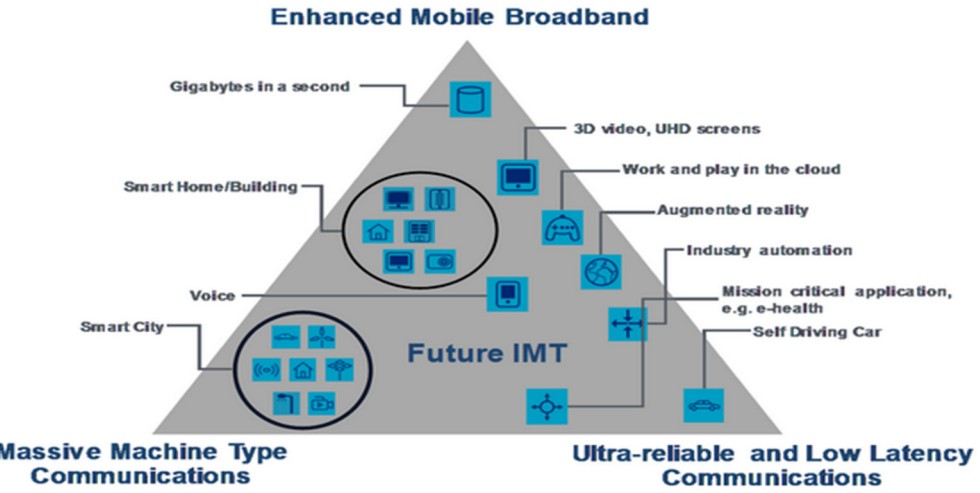

**Figure 1.** 5G Usage Scenario.

In the case of 5G-mmWave, the typical deployment uses a frequency carrier at 26–28 GHz, with a bandwidth of 100 MHz and 400 MHz. Table 1 shows the calculation for the peak data rate using (1) with a technical specification of $8 \times 8$ MIMO at four modulation scheme.

**Table 1.** Peak data rate vs. Modulation for 26/28 GHz.

| Freq Band 26/28 GHz | Modulation | | | |
|---|---|---|---|---|
| | QPSK | 16 QAM | 64 QAM | 256 QAM |
| 100 MHz | 2.16 Gbps | 4.31 Gbps | 6.47 Gbps | 8.62 Gbps |
| 400 MHz | 8.62 Gbps | 17.24 Gbps | 25.86 Gbps | 34.48 Gbps |

Source: Authors' calculations.

### 2.2. 5G Private Network for Industrial Areas

The 5G private network, also known as the 5G local network or 5G non-public network, is a local area network that underlies 5G technology, having all the characteristics of a 5G network, including a lower latency and faster speeds. It is believed that 5G private networks will provide better overall connectivity, with a variety of benefits and enhanced services for users of the limited geography network. The 5G private network is likely to become the best choice for many of the world's corporations, especially for industrial sectors, such as manufacturing plants, remote offices, and terminals. Figure 2 illustrates the 5G private network [16].

One significant aspect of 5G private networks is their ability to improve automation in the industry by using certain advanced technological platforms, such as artificial intelligence technologies (AI), Internet of Things (IoT), and machine learning [17]. Thus, 5G private networks may become essential to creating smart factories, including industrial campus network solutions, smart industrial areas, and smart manufacturing companies [18].

One function that sets 5G private networks apart from other services is that they allow a company to work as a mobile private 5G network operator (MPNO). With this function, the company can set up its 5G private network infrastructure and have absolute authority over its private network, including its configuration, procedures, technical performance, and other related network management issues. Alternatively, the company can subscribe to public service operators who also run public 5G private network services. Under this conception, the private network is only part of the business model run by the 5G service operator, in which the company is simply classified as a corporate user [19].

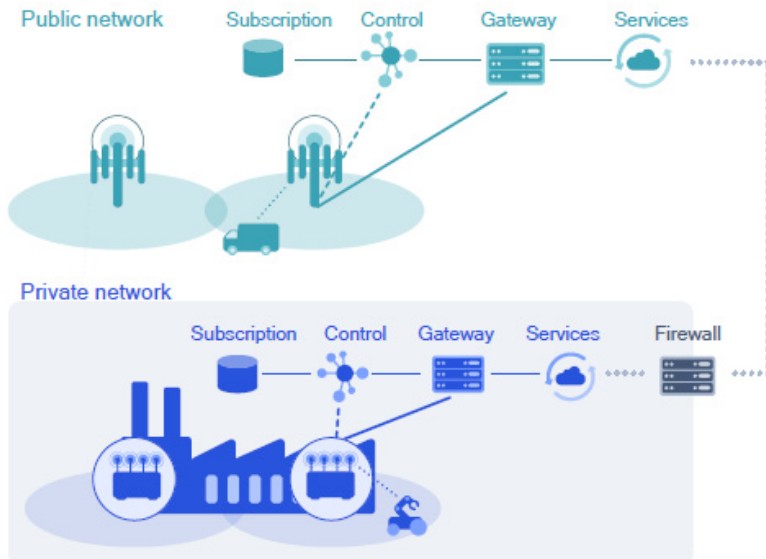

**Figure 2.** 5G private network.

*2.3. 5G Spectrum Usage Fee*

Previous studies of spectrum usage fees include [4–6], which examine the formulation of spectrum usage fees for service operators (2G/3G) in Taiwan and Indonesia. The formulas used by these nations vary depending on various factors, including competition among service operators, geographical condition, population, bandwidth, and expected income from spectrum usage fees for the government. The goals of each nation are the same—namely to formulate a fair and technology-neutral fee model to enhance the efficiency of spectrum utilization. The framework was developed based on frequency bands below 3 GHz, operated on narrow bandwidths of up to 20 MHz for the 4G network.

5G provides three different usage scenarios that require simultaneous access to low, mid, and high bands to meet 5G capacity, latency, coverage, and quality requirements to comply with advanced spectrum range use cases [20,21]. Therefore, the formulation of the spectrum usage fee may have a different focus for different bands. Recent research has begun to conduct spectrum valuation and pricing of private LTE and 5G networks at the 3.5 GHz frequency band [7–10].

*2.4. 5G-mmWave Private Network Engineering Economic Model*

From the perspective of the engineering model, the primary focus of deployment in the 5G-mmWave private network's rollout is network densification, especially as mmWave bands with worse propagation properties become more prevalent [22]. The spectrum mmWave has the potential to offer multi-Gbps data rates at a lower marginal cost than earlier technologies due to the potential allocation of bandwidth with significantly more width at these frequencies. mmWave bands will be used for smaller coverage, such as private networks in industrial areas, resulting in different deployment economics than previous generations [23].

Furthermore, various studies conducted using mmWave and real-world deployments of the mmWave spectrum (26, 28, and 73 GHz) show that cell coverage using these frequencies will have a limited radius of 100–200 m [24,25]. To calculate the link budget assumption data, the engineering model uses the 3GPP 36,901 propagation model. This concept was first applied to frequency ranges of 2–6 GHz before being expanded to 0.5–100 GHz for urban macrocells (UMa), urban microcells (UMi), indoor hotspots (InH), and rural macrocells (RMa) [26].

There may be coverage problems in Pulogadung and KBN industrial areas covered by mmWave gNB during the initial deployment phase of the 5G-mmWave private network.

Due to the weakness of reflected and diffracted signals, the propagation will rely heavily on LoS coverage. Indoor users expect to be served by mmWave gNB in the 5G-mmWave private network, which will provide exceptionally high capacity and the necessary resources for higher data rates. Increased penetration loss at high frequencies is one of the challenges in this context [27].

Because of penetration loss and atmospheric loss, a highly directional antenna is recommended for a 5G-mmWave private network to compensate for additional losses due to path loss in the industrial area [27].

Beamforming will ensure reliable links for deploying 5G-mmWave private networks when combined with high gain or large array antennas at the transmitter and receiver ends [28].

The ability of 5G-mmWave private networks to handle a large number of IoT devices in the industrial area is a crucial design goal. The development of cellular networks with extreme densification and additional spectrums are two possible solutions to meet these demands [29].

From an economic perspective, the cost of mmWave band spectrum densification in 5G-mmWave private network infrastructure is heavily influenced by the required CAPEX and OPEX [22,30,31]. Because network infrastructure has very high fixed delivery capital expenditure (CAPEX) and operational expenditure (OPEX), it is heavily influenced by scale economies and population density [32]. CAPEX is an investment or outlay made to buy or improve physical assets, such as land, buildings, and equipment, or to expand activities. In this study, the total cost of base station building, base station installation and spectrum usage fees are calculated in terms of radio access network (RAN) CAPEX [33]. In telecommunications performance, OPEX includes spending for operation and maintenance charges, as well as other telecommunications costs.

## 3. Indonesian Profile

### 3.1. Mobile Ecosystem and Spectrum Regulatory Policy

With a population of approximately 270.2 million, Indonesia is the fourth most populous country in the world, with a land area of almost two million square kilometers spread over 13,000 inhabited islands (out of 17,000). However, more than 56.10% of the land area is concentrated on Java Island, and the country's population is not equally distributed over its land mass [34]. Java Island, which contributes 58.70% of the country's GDP, dominated the spatial structure of the Indonesian economy in Quarter 1 2021, followed by Sumatera Island at 21.54%, Kalimantan Island at 8.05%, Sulawesi Island at 6.52%, and Bali and Nusa Tenggara at 2.75%. The Maluku Island and Papua provincial groups accounted for the lowest contribution at 2.44% [35].

Concerning internet users in Indonesia, Figure 3 shows the percentage of the Indonesian population with internet access. Java Island contains more than half (56.4%) of Indonesia's internet users, followed by Sumatra Island (22.1%), Bali-Nusa Tenggara (5.2%), Kalimantan (6.3%), Sulawesi (7%), and Maluku-Papua (3%) [36].

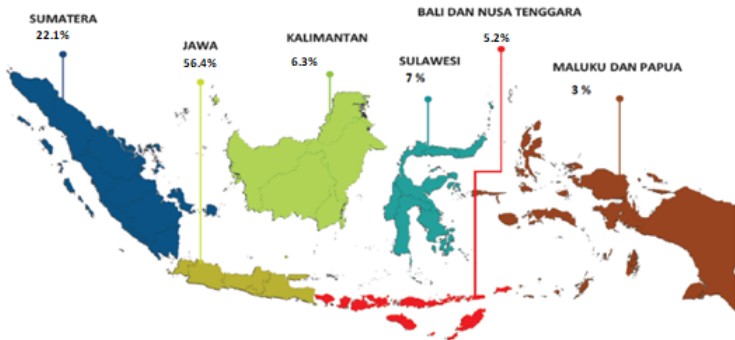

**Figure 3.** Contribution of internet users by region from all users [36].

In 2021, there were around 345.3 million mobile phone subscribers (market penetration 125.6%) [37], many of whom had more than one mobile subscription. The country has 202.6 million internet subscribers, 96.4% of whom prefer to access the internet via mobile devices. Indonesia is currently served by six public service operators deploying the 2G, 3G, and 4G network infrastructure. Figure 4 shows the proportion of each service operator's subscribers in Indonesia [38].

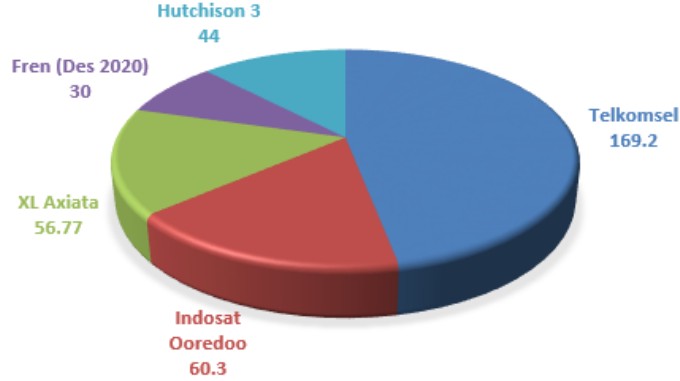

**Figure 4.** The proportion of each service operator's subscribers in Indonesia (million) [38].

From 2017 to 2021, some Indonesian service operators developed trial tests for 5G band candidates as part of their deployment plan [38]. Most of the trial tests used the mmWave frequency of 28 GHz (24.25–29.5 GHz) [39]. An experiment with 3.5 GHz (3.3–4.2 GHz) was also carried out in an indoor area. In 2021, Telkomsel and Indosat launched the first commercial 5G enterprise in Indonesia [40].

In Indonesia, the spectrum usage fee policy is run by the Ministry of Communication and Information in accordance with the provisions of the applicable laws. Under the law product of [2], any service operator running the network infrastructure on a certain spectrum has been obliged to pay the spectrum usage fee as state income. Such government income is categorized as non-tax income and is expected to gradually increase every year. The fee is formulated with the motivation to promote the effective and efficient use of the frequency spectrum and to fairly contribute to the country's overall development plan.

Figure 5 shows the amount of spectrum usage fees paid by the big three service operators in Indonesia in 2010–2020. The trend shows a growing income that can be interpreted as Indonesian service operators massively deploying 3G and 4G technology during this period. The amount of the paid spectrum fee corresponds directly to the capacity infrastructure built by the service operators [41].

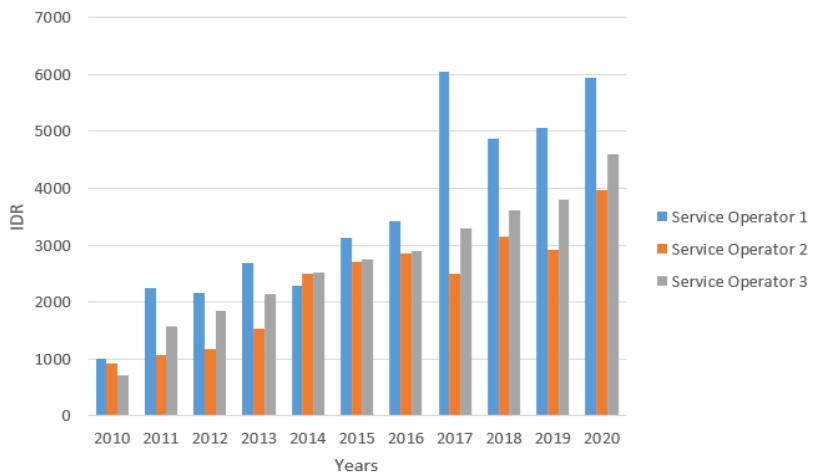

**Figure 5.** Spectrum usage fees of the three biggest service operators in Indonesia (IDR Million) [41–44].

*3.2. The Indonesian Industry*

In Indonesia, the manufacturing sector plays an important role in national economic growth, contributing 17.34% to the national GDP in the second quarter of 2021 and ranking fifth among the country members of the G20 [45].

The government has established the industrial target under the national program Making Indonesia 4.0 [36]. The final target is to make Indonesia one of the world's top 10 largest economies by 2030 and to create export-driven economic growth. Another target is to return the industrial net export rate to a 10% contribution to GDP, doubling the workers productivity rate over the workers costs and allocating 2% of GDP to R&D and technology innovation fields.

To achieve such targets, the government is boosting digital transformation in the industrial sectors with several initiatives, following Making Indonesia 4.0 [36]. The aim is to integrate primary innovation into the production value chain in the areas of technology, information, communications, connectivity, and hardware automation in the manufacturing sectors. By linking such an initiative with the 5G-mmWave private network deployment plan, it would be a significant action to implement 5G-mmWave private networks in manufacturing companies.

Indonesia has hundreds of manufacturing companies in various industrial areas spread over Indonesia's main islands. As listed in Table 2 [46], there were 141 industrial areas in 2019, including one under construction. The majority of the industrial areas are located on Java Island, accounting for 73 locations, while the least industrialized area is Nusa Tenggara, which has only one location.

**Table 2.** The number of operational industrial areas in 2019 [46].

| No | Islands | Numbers of Industrial Areas | Area (Ha) | Percentages |
|----|---------|-----------------------------|-----------|-------------|
| 1 | Java | 73 | 39,444.37 | 60.7% |
| 2 | Kalimantan | 19 | 7301.46 | 11.2% |
| 3 | Sulawesi | 6 | 5502.00 | 8.5% |
| 4 | Sumatera | 40 | 11,969.40 | 18.4% |
| 5 | Maluku and Papua | 2 | 600 | 0.9% |
| 6 | Nusa Tenggara | 1 | 191 | 0.3% |
| | Total | 141 | 65,008.23 | 100.0% |

Making Indonesia 4.0 prioritizes the following five industrial sectors: food and beverage, textiles, automotive, chemical, and electronics. These sectors are spread over several industrial areas [47] and have become the target areas for several reasons. For example, the food industry holds the biggest share, with a total investment realization (both foreign and domestic) of IDR 302.8 trillion. This is followed by the sectors of basic metal, metal goods, machines, and electronics, worth IDR 299.0 trillion. The chemical sector is at third place, worth IDR 285.5 trillion. While the textile industry accounts for only IDR 58.3 trillion, its realization would bring more workers than other industries.

## 4. Evaluation of Current Formula for Spectrum Usage Fee

*4.1. Origin of The Current Formula*

The document ITU-R SM.2012-5 (06/2016) provides an analytical model to formulate the spectrum usage fee on the basis of specific incentives designed to promote the efficient use of the spectrum in any country [48]. The formula is developed based on the concept that there may be different requirements for valuating the spectrum that reflect more than administrative convenience. However, the main paradigm of the model is to keep improving the efficiency of spectrum utilization.

Indonesia has adapted and modified this model with the country's specific administrative incentives price (AIP). Figure 6 shows how Indonesia's current spectrum usage fee adapts the ITU-R SM.2012-5 (06/2016), resulting in (5).

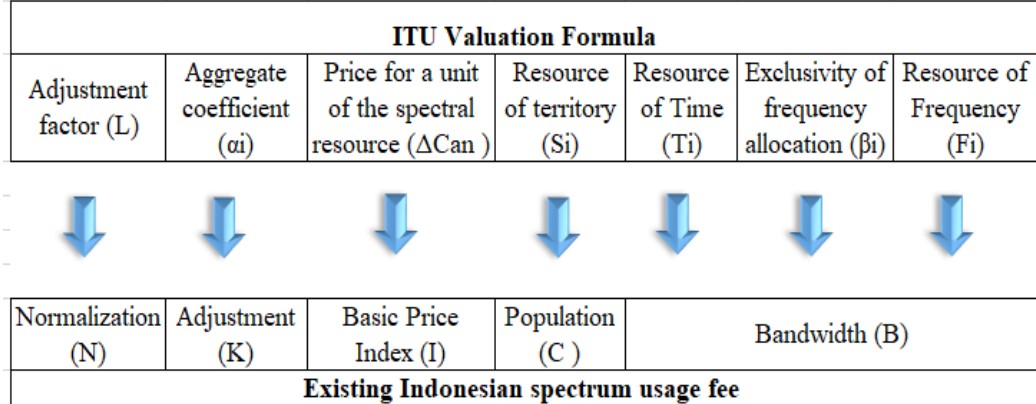

**Figure 6.** Spectrum valuation formula from ITU-R SM.2012-5 (06/2016), adapted from the current Indonesia spectrum usage fee.

The sequential narration for Figure 6 is as follows:

(1)   Determine the total amount of the annual payments for the spectral resource ($C_{an}$)

$$C_{an} = C1 + C2 - I_{an} \tag{2}$$

where *C1* is the expense of national and international spectrum management, *C2* is state net income (optional), and $I_{an}$ is the sum of the annual inspection charge.

(2)   Determine the value of the spectral resources ($W_i$)

$$W_i = \alpha_i \times \beta_i \times (F_i \times S_i \times T_i) \tag{3}$$

where $F_i$ is the resource factor of frequency, $S_i$ is the resource of territory, $T_i$ is the resource of time, $\alpha_i$ is an aggregate coefficient that takes a variety of weighting variables into account, such as commercial, social, and operational factors, and $\beta_i$ is the weighting coefficient that defines the exclusivity of frequency allocation.

(3)   Determine the annual price for a unit of the spectral resource ($\Delta C_{an}$), with the unit being the country's currency per MHz.km$^2$

$$\Delta C_{an} = L \times \left( \frac{C_{an}}{W} \right) \tag{4}$$

where *L* is an adjustment factor that can be determined by the government for the next fiscal year.

Figure 6 also shows that when adopted by the Indonesian government, each factor defined by the International Telecommunication Union (ITU) is translated into five factors that structure the Indonesian spectrum usage fee. Equation (5) states the current formula, and its factors are explained in Table 3 [2].

$$\text{Indonesian spectrum usage fee} = N \times K \times I \times C \times B \tag{5}$$

The government of Indonesia has set the value of $N \times K$ to 28.36 for all bands [3], while the value of *I* is shown in Table 4. The table indicates that when the frequency becomes higher, the price unit becomes cheaper. Concerning the resource factor of territory, Indonesia applies the concept of nationwide population (kilo population) as a measure for the *C* value. However, as noted by one researcher [7,8], the nationwide population might not accurately assess the valuation of 5G-mmWave private network licenses, as these will be used in an industrial area.

**Table 3.** The current formula for Indonesian spectrum usage fees [2].

| Factor | Explanation |
| --- | --- |
| Normalization ($N$) | The N value is adjusted annually using the value of the consumer price index (CPI). To determine the N value, the CPI data are required for the one and two preceding years or are variable in the form of d ($n − 1$). The Indonesian government uses this factor to maintain a stable amount of non-tax state income. |
| Adjustment ($K$) | This is an adjustment factor obtained based on the economic value of the frequency band, including the type of service, the benefits received, and the service area. |
| Basic price index ($I$) | Basic price index for radio frequency bands (IDR/MHz). |
| Bandwidth ($B$) | The amount of radio frequency bandwidth allocated according to the specified spectrum band, including the guard band or bandwidth that cannot be used by other users. |
| Population ($C$) | The total population in a service area. The unit $C$ is the kilo population (per 1000). |

**Table 4.** The Indonesian basic price index value (I) for radio frequency bands [2].

| Frequency Range (MHz) | Units | Basic Price Index (IDR) |
| --- | --- | --- |
| 3400–4500 | per MHz | IDR 4508 |
| 4500–5000 | per MHz | IDR 4393 |
| 5000–8500 | per MHz | IDR 3811 |
| 8500–11,700 | per MHz | IDR 3461 |
| 11,700–12,750 | per MHz | IDR 3367 |
| 12,750–15,400 | per MHz | IDR 3160 |
| 15,400–22,000 | per MHz | IDR 2769 |
| 22,000–31,300 | per MHz | IDR 2383 |
| 31,300–52,600 | per MHz | IDR 1814 |

*4.2. Problem of the Current Formula Used for 5G-mmWave*

In our previous work [41], we simulated the amount of the spectrum usage fee for 5G-mmWave that should be paid by the three biggest Indonesian service operators using the current formula. Figure 7 shows that when the Indonesian service operators use a bandwidth of more than 100 MHz; their profit tends to be dynamically under turbulence, with only Telkomsel (the biggest service operator in Indonesia) seeing positive growth up to the eighth year but then declining significantly.

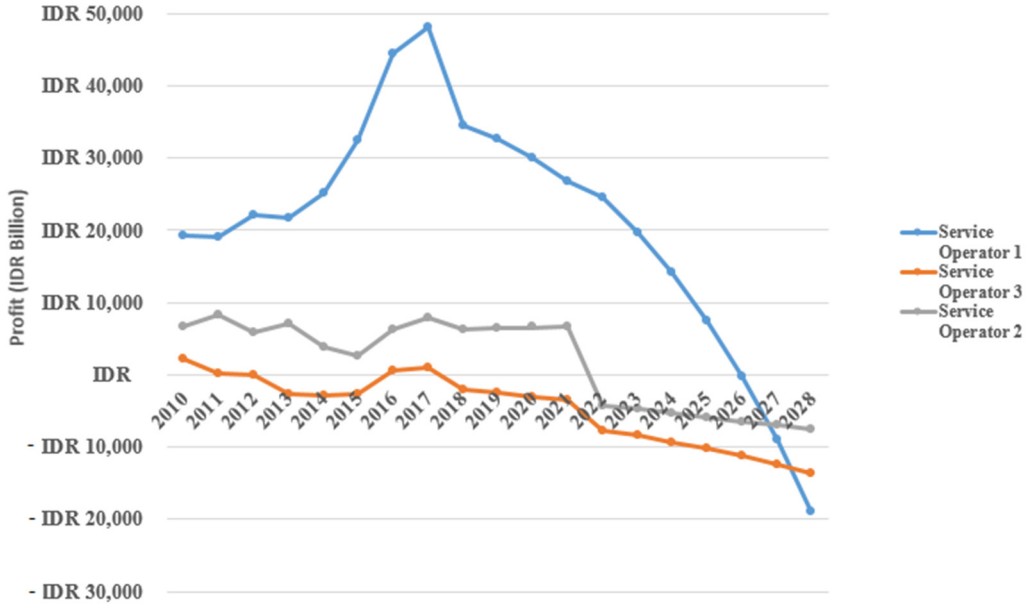

**Figure 7.** The trend of the service operators' profit as a result of 5G implementation in Indonesia [41].

Based on research by [49], if the service operators have a spectrum usage fee that is greater than their gross income, their business will become unsustainable.

By using the current formula, some Indonesian service operators will lose profit even from the start of the 5G implementation. This phenomenon is due to the fact that the current formula uses the national population instead of the specific number of service subscribers. However, because the 5G private network provides limited coverage, it can serve only a certain part of the population. In short, the current spectrum usage fee formula is no longer feasible for the 5G private network, making the fee very expensive.

This paper presents proposed alternative representative measures to assess the location value with limited geography for 5G-mmWave private network licenses. The opportunities lie in the fact that not only do mobile applications today and in the future need wider bandwidth, but also that their coverage area can be developed within a small area, rather than on a national scale, and used by companies in industrial areas.

## 5. Methods and Proposed New Formula

### 5.1. Methods

This study proposes a new formula for spectrum usage fees, using the ITU-R SM.2012-5 (06/2016) framework and an industry reference index, called the Indonesia Industry Readiness Index 4.0 (INDI 4.0) score, for five priority industrial sectors. With a limited geographic network or 5G-mmWave private network, the reference index is taken into account when developing new formulas. The score was obtained by surveying 303 companies from five priority industrial sectors. The survey was carried out in 2019–2020.

After the new formula for the spectrum usage fee is ascertained, the next step is to calculate the CAPEX and OPEX costs for 5G-mmWave private network deployments located in two industrial areas (Pulogadung, KBN) using the Engineering-Economic Model based on the new spectrum usage fee formula from the previous output.

The results of the Engineering-Economic Model for 5G-mmWave private networks in industrial areas is then used for the calculation of Indonesian Input-Output (I–O) analysis. The dimensions of the Input-Output (I–O) matrix is 17 × 17. This simulation estimates the contribution of 5G-mmWave private network implementation to the output of the national economy, where GDP is a part of the economic output. Figure 8 shows the new spectrum usage fee model adopted in this study.

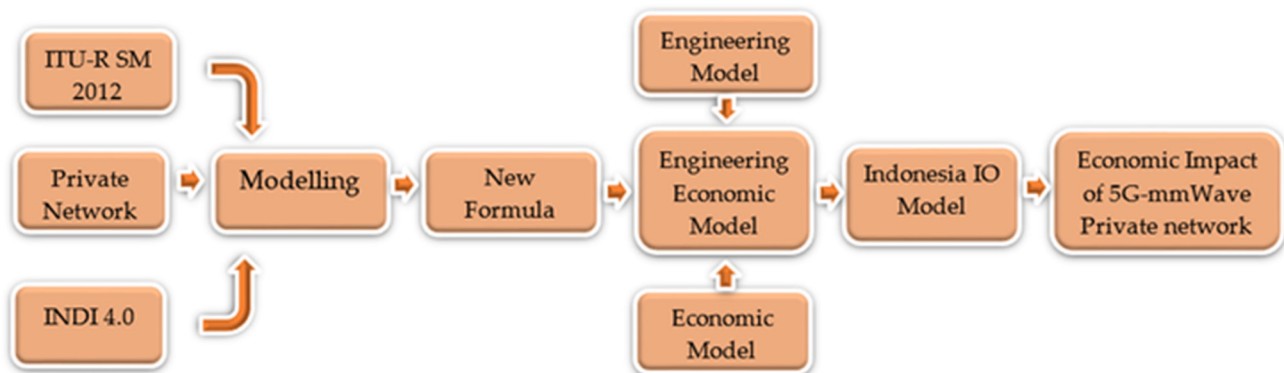

**Figure 8.** Indonesia spectrum usage fee model.

### 5.2. Proposed New Formula for Spectrum Usage Fees

In this paper, we propose a framework leading to the formulation of a new spectrum usage fee for 5G-mmWave private network implementation in the industrial areas of Indonesia. We developed the proposed new formula mainly by adopting the INDI 4.0 scores for Indonesia's five priority industrial sectors. After considering such factors, we also took into account the number of workers in the industry and combined them with the ITU-R SM.2012-5 (06/2016) framework.

### 5.2.1. Significance of Industrial Reference Index

We considered the industrial reference index to reflect the readiness of the industrial sector to adopt technology in the Industrial 4.0 era. To calculate the readiness of companies for Industry 4.0, the Indonesian government formed the INDI 4.0 score to set forward reference criteria. The score was obtained by surveying 303 companies in the five priority industrial sectors [50].

As seen in Table 5, the score is calculated using certain measurements, including leadership policies as well as the workers and work culture in the company, relevant to Industry 4.0. The score also measures products and services in relation to the technology used by the industrial companies in their digital transformation, including the operation of cyber security platforms, IoT, and other supporting technologies. A final measurement concerns factory operations, involving the performance of data storage and sharing, supply chain and intelligent logistics, autonomous processes, and intelligent maintenance systems.

**Table 5.** $C_{INDI}$ score for the five priority industrial sectors scores [50].

| No | Sectors | Numbers | Score on the Measurement | | | | | $C_{INDI}$ |
|----|---------|---------|-----|-----|-----|-----|-----|-------|
| | | | **(1)** | **(2)** | **(3)** | **(4)** | **(5)** | |
| 1 | Textile and clothing | 10 | 2.7 | 2.5 | 2.8 | 2.3 | 2.3 | 1.16 |
| 2 | Food and drink | 39 | 2.5 | 2.51 | 2.6 | 2.3 | 2.42 | 1.14 |
| 3 | Chemistry | 30 | 2.34 | 2.33 | 2.4 | 2.1 | 2.33 | 1.07 |
| 4 | Electronic | 28 | 1.47 | 1.89 | 2.2 | 1.6 | 1.9 | 0.85 |
| 5 | Automotive | 196 | 1.35 | 1.83 | 2 | 1.5 | 1.83 | 0.79 |

Notes: (1) = measurement of leadership policies; (2) = measurement of workers and work culture; (3) = measurement of product and services; (4) = measurement of technology; (5) = measurement of factory operations.

Table 5 presents the results for each measurement and the final INDI score. The highest INDI score (1.16) is in the textile sector, and the lowest score, with an average of 0.79, is in the automotive industry.

### 5.2.2. Considering the Number of Workers in the Industry

The second consideration of the new formula is the number of workers in the industry. This formula uses the INDI 4.0 average because the number of workers might not be proportional among different industries in the five priority sectors. Hence, it is important to consider the number of workers relative to the industry area and the total number of workers in that province.

### 5.2.3. Proposed New Formula

Based on the two conceptions above, we propose a new spectrum usage fee formula, as in (6). We highlight the factor of $C_{Alt.}$ as the function of the INDI score ($C_{INDI}$), the number of workers in the industrial area ($C_{EDIA}$), the total number of workers in the province where the industrial area is located ($C_{EDPROP}$), and the size of the industrial area ($A_{IA}$).

$$\text{Proposed New Formula for Spectrum Usage Fee} = N \times K \times I \times C \times B \times C_{Alt} \quad (6)$$

$$C_{Alt} = \left( \frac{C_{EDIA}}{C_{INDI} \times C_{EDPROP}} \right) \times \left( \frac{1 \text{ km}}{A_{iA}} \right) \quad (7)$$

where $C_{INDI}$ is the INDI 4.0 score (as in Table 5), $C_{EDIA}$ is the number of workers in the industrial area, $C_{EDPROP}$ is the total number of workers in the province where the industrial area is located, and $A_{IA}$ is the size of the industry area.

The equation indicates that the higher the $C_{INDI}$, the lower the spectrum usage fee, and the wider the area, the lower the fee. This relation makes sense for the industry because the INDI score acts as an incentive value for the relevant industry. The better they

perform, the more benefit they receive and the lower the obligation to pay the fee by the service operators.

## 6. Results

In this section, we demonstrate proof of concept by testing the new formula for calculating the spectrum usage fee in Indonesian industrial areas. The calculation is for a 10-year license fee in the mmWave spectrum band with a bandwidth of 100 MHz. To ease reading, all the values have been rounded without displaying the decimal values. We take study cases focusing on the Jakarta region in two industrial areas: Pulogadung and KBN. Jakarta is the capital city of Indonesia, which may be considered an ideal example of 5G-mmWave private network vertical industry implementation. The Pulogadung industrial area is in Jakarta and is jointly possessed by the Republic of Indonesia and the Jakarta provincial government. It covers 500 ha (5 km$^2$) and is home to 400 companies, while KBN is an Indonesian state-owned enterprise engaged in managing industrial areas located in Jakarta that cover 800 ha (8 km$^2$). In 2020, the government of Indonesia held 73.15% of the company, while the Jakarta provincial government held the rest.

The data about both industrial areas are shown in Table 6, in which columns) (3,4) are the names of the areas and rows (2–5) list basic information about them. We calculate the spectrum usage fee using the new formula, as shown in rows (6–10) of Table 6. These rows show the results for each type of industry—namely, chemistry, food and drink, automotive, textiles and clothing, and electronic. The total sum of all the industrial types is indicated in row (11) of Table 6.

**Table 6.** Spectrum usage fee comparison between the current and proposed new formula in five priority industries (IDR) and 5G-mmWave private network deployment costs for the 10-year duration of the spectrum usage fee.

| | (1) | (2) | (3) | (4) |
|---|---|---|---|---|
| (1) | | | **Pulogadung** | **KBN** |
| (2) | | Number of workers in industrial area ($C_{EDIA}$) | 65,000 | 80,000 |
| (3) | BASIC INFORMATION | Area ($A_{IA}$) in km$^2$ | 5 | 8 |
| (4) | | Province | Jakarta | Jakarta |
| (5) | | Number of workers in whole province of Jakarta ($C_{EDPROP}$) | 145,000 | 145,000 |
| (6) | | Chemistry sector | IDR 740 million | IDR 714 million |
| (7) | SPECTRUM USAGE FEE with new formula per Industrial sector for the 10-year duration | Food and Drink sector | IDR 692 million | IDR 668 million |
| (8) | | Automotive sector | IDR 999 million | IDR 965 million |
| (9) | | Textiles sector | IDR 681 million | IDR 657 million |
| (10) | | Electronic sector | IDR 934 million | IDR 901 million |
| (11) | TOTAL SPECTRUM USAGE FEE with new formula for the 10-year duration | (a) Total new formula: sum of all priority sectors | IDR 4046 million | IDR 3905 million |
| (12) | TOTAL SPECTRUM USAGE FEE with current formula for the 10-year duration | (b) Spectrum usage fee using the current formula | IDR 4392 million | IDR 5406 millions |
| (13) | | (c) Differences = (a)—(b) | −IDR 347 million | −IDR 1500 millions |
| (14) | DIFFERENCES | Percentage decreased between new formula and current formula | 8% loss | 28% loss |
| (15) | | CAPEX and OPEX Cost to deploy 5G-mmWave private network infrastructure in the industrial area | IDR 193,911 million | IDR 267,126 million |

The explanation below demonstrates how to obtain such spectrum usage fee values. The calculation example is for the chemistry sector in the industrial area of Pulogadung.

To acquire the value of the spectrum usage fee using the current formula (Table 6—row (12) column (3))

| | |
|---|---|
| N × K | = 28.36 |
| Index Price (I) | = IDR 2383 |
| Kilo population (C) | = 65 |
| Bandwidth (B) | = 100 MHz |

Therefore, the current spectrum usage fee value is
= N × K × I × C × B
= 28.36 × IDR 2383 × 65 × 100
= IDR 439,221,848 per year or IDR 4,392,218,48 for 10 years
(Rounded to IDR 4392 million for the 10-year fee)
To acquire the value of the spectrum usage fee using the new formula for the chemistry industry (Table 6—row (6) column (3))

| | |
|---|---|
| N × K | = 28.36 |
| Index Price (I) | = 2383 |
| Kilo population (C) | = 65 |
| Bandwidth (B) | = 100 MHz |
| $C_{Alt}$ | = 0.16844 |

Therefore, the current spectrum usage fee value for Chemistry sector in Pulogadung is
= N × K × I × C × $C_{Alt}$ chemistry × B
= 28.36 × IDR 2383 × 65 × 0.16844 × 100
= IDR 73,983,868 per year or IDR 739,838,681 for 10 years
(Rounded to 740 million for the 10-year fee)

The sum of the 10-year spectrum usage fees produced using the proposed new formula for all industry sectors in Pulogadung is listed in Table 6—row (11) column (3), gives the value of IDR 4,046 million. Comparing the new value with that of the current formula yields a difference of about IDR 347 million (Table 6—row (13) column (3)), which is approximately 8% lower than the current formula.

## 7. Discussion

The results in Table 6 clearly indicate that the proposed new formula will always produce a lower value than the current formula for the spectrum usage fee. Such a value would have a positive impact on the industry as service operators would pay a cheaper price, making the economic cost more efficient. However, it is a common understanding that the government expects state income from commercializing new technology. A lower value means that the country loses some potential income from the spectrum usage fee. How then, can this loss in state income be compensated for? We posit that it can be compensated for by the sector's multiplier effect at the national level. In other words, despite such a loss, the lower value would result in a far greater positive economic impact at the national level.

To prove our argument, we develop the I–O model to show how 5G-mmWave private network implementation in Pulogadung and KBN industrial areas will produce an economic impact greater than the lost state income. This economic impact will also compensate for the difference in spectrum usage fees.

### 7.1. Input-Output (I–O) Model

The I–O model is often used in the analysis of industrial systems or macroeconomic systems to examine the structure of intersectoral linkages. The I–O model can show the size of the flow of intersectoral linkages in an economy. The relationship between the input arrangement and the output distribution is the basic theory underlying the model.

The I–O model describes the transaction stream between sectors in which each sector provides a certain 'output' while also consuming 'inputs' from other sectors. In our case, the 5G-mmWave private network deployment CAPEX and OPEX cost serves as the input, and the economic impact as the output.

In the I–O model, the parameter of 'multiplier' calculates the ratio of output changes in equilibrium as a result of a change in final demand. Thus, the multiplier calculates the

overall change in the economy as a result of a unit change in final demand. Changes in final demand may result from private consumption, government spending, investment, and export. A change in output will be greater than a change in final demand as a result of production linkages. For example, if the information and communication technology (ICT) sector's final demand (e.g., broadcasting and programming services, film and sound recordings, telecommunication services, and computer and information technology consulting services) rises by 10 unit values of money, the economy's output will rise by more than 10 unit values of money or by the multiplier coefficient of these sectors.

### 7.2. Economic Impact of 5G-mmWave Private Network in Industrial Areas

Translating the I–O model to our case, the input is the total CAPEX and OPEX cost of deploying 5G-mmWave private networks that will impact economic output.

To do so, we first calculate the total CAPEX and OPEX cost in the industrial area over a 10-year period. The total CAPEX and OPEX cost of deploying 5G-mmWave private networks is determined by the following factors: base station core network construction, installation, and maintenance, optical fiber installation and maintenance, and spectrum usage fees. We then estimate a 10-year capital expenditure schedule from 2021 to 2031, with the assumption that all costs are fixed costs [51]. In Table 7, row (15) defines the final total CAPEX and OPEX cost of deploying 5G-mmWave private networks in each industrial area. The values result from our calculations applying the engineering economic method [24] to each industrial area.

The input for the I–O model is the cost of 5G-mmWave private network deployment We assume that the impact can be ascertained in the 17 sectors officially defined by the Indonesian government [46,52]. These 17 sectors should be affected by the input of the 5G-mmWave private network deployment cost listed in Table 7, column (1) rows (2–18). The sectors range from agriculture to health services, taking in various miscellaneous sectors.

Table 7, column (2) defines the value of the multiplier. The multiplier is the proportional increase or decrease in the final income that results from an injection or withdrawal of spending. Table 7, columns (3) and (4) list the results of the I–O model that yield the output, or economic impact, due to the input of the 5G-mmWave private network deployment cost. Table 7, columns (3) and (4) indicate the impact on the 17 sectors when the 5G-mmWave private network is deployed in the Pulogadung and KBN industrial areas, respectively. The overall sum of these sectors is listed in row (19).

**Table 7.** Multiplier coefficient and economic impact from 5G-mmWave private network deployment investment (Million IDR).

|  | (1) | (2) | (3) | (4) |
|---|---|---|---|---|
| (1) | Sectors | Multiplier Coefficient | Output (National Economic Impact) | Output (National Economic Impact) |
| (2) | Agriculture forestry and fisheries | 1.28 | IDR 2274 million | IDR 3123 million |
| (3) | Mining and quarrying | 1.46 | IDR 5478 million | IDR 7547 million |
| (4) | Processing industry | 1.73 | IDR 14,487 million | IDR 19956 million |
| (5) | Electricity and gas procurement | 2.95 | IDR 17,641 million | IDR 24,301 million |
| (6) | Water supply, waste management, waste and recycling | 1.63 | IDR 79 million | IDR 109 million |
| (7) | Construction | 1.82 | IDR 1378 million | IDR 1898 million |
| (8) | Wholesale and retail trade, car and motorcycle repair | 1.43 | IDR 4528 million | IDR 6237 million |
| (9) | Transportation and warehousing | 1.78 | IDR 4683 million | IDR 6451 million |
| (10) | Provision of accommodation and food | 1.75 | IDR 1559 million | IDR 2148 million |
| (11) | Information and communication | 1.59 | IDR 224,419 million | IDR 309,153 million |
| (12) | Financial services and insurance | 1.39 | IDR 5398 million | IDR 7436 million |
| (13) | Real estate | 1.36 | IDR 2956 million | IDR 4073 million |
| (14) | Company services | 1.59 | IDR 13,656 million | IDR 18,812 million |
| (15) | Government administration, defense, and social security | 1.70 | IDR 4231 million | IDR 5829 million |
| (16) | Education services | 1.51 | IDR 271 million | IDR 373 million |
| (17) | Health services and social activities | 1.73 | IDR 129 million | IDR 178 million |
| (18) | Other services | 1.56 | IDR 5217 million | IDR 7187 million |
| (19) | Sum |  | IDR 308,384 million or 0.244% economic output contribution | IDR 424,820 million or 0.336% economic output contribution |

Thus, Table 7, row (19) shows that the 5G-mmWave private network in the Jakarta industrial area will produce an output of 308,384 million IDR for the Pulogadung industrial area and 424,820 million for the KBN industrial area. These values are the increase in the contribution to the output of the Indonesian economy as an effect of the multiplier.

Table 6, column (3), row (13), shows that implementing the new formula to calculate the spectrum usage fee in the Pulogadung area would result in a loss of IDR 347 million in state income. However, it will be compensated for by contributing IDR 308,384 million to the Indonesian national economic output, accounting for a total economic output increase of 0.244% over a 10-year period.

By detailing the 17 sectors, we see that the 5G-mmWave private network deployment cost would benefit each sector. For instance, the information and communication section (row (11), column (2)) has a multiplier of 1.59. The value indicates that if there is a final demand of IDR 1 million, the output of this sector will increase by IDR 1.59 million. Translated into this case, this means that the 5G-mmWave private network deployment in the Pulogadung area will create a national economic impact of IDR 224,419 million in the information and communications sector. As another example, the health services and social activities sector (row (17), column (2–4)) has a multiplier of 1.73, and the national economic value of the output is approximately IDR 129 million and IDR 178 million due to 5G-mmWave private network deployment in the Pulogadung and KBN areas, respectively.

Finally, it should be noted that countries with advanced telecommunications are not solely concerned with an annual increase in state income. Optimizing the mmWave frequency for the 5G-mmWave private network with a broader bandwidth will provide the country with even better value for the multiplier coefficient supporting Industry 4.0 and promoting digital economic growth and national digital transformation.

*7.3. Advantages of the New Formula*

The industry clearly benefits from the spectrum usage fee reduction offered by the new formula. Some of the advantages for regulators are as follows:

(1)     Regulators treat the reduced fee as a subsidy using the readiness of each industry according to INDI 4.0. The spectrum usage fee value of the new formula is lower than that of the old formula, which can help service operators survive. Various 5G-mmWave local micro-operators or 5G-mmWave private network service operators are not directly covered by the Indonesian regulator. There is therefore a demand for a new spectrum authorization design to allow new micro-operators to participate in 5G. The regulator grants exclusive licensing for service operators to provide long-term, usually 10 years, for a spectrum access license.

(2)     The new formula supports harmony and mutual support between industrialization and digital transformation. The Ministry of Communication and Informatics, as the regulator in the field of telecommunications, must work closely with other ministries to determine an overall general roadmap for the success of new technologies because the 5G-mmWave private network covers a wide range of industries and involves close coordination between various government departments. Without coordinated standards and policies, the true benefits of 5G-mmWave private networks will not be realized. Such collaboration will help drive the innovation ecosystem with customized offerings that meet the specific needs of the industry.

(3)     Because the coverage of this private network is limited to specific sites with a restricted coverage area, the 5G-mmWave private network usage scenarios will create immense opportunities to increase service operators' infrastructure and provide more significant economic benefits. The ongoing technological expansion of 5G-mmWave private networks has opened up opportunities for various new spectrum applications. Although frequently making spectrum use more efficient, these expansions have pushed significant interest and demand for the limited spectrum resource.

(4)     It will be easier to reuse these frequencies in other locations. Indonesia, which has advanced telecommunications technology, not only pays attention to increasing non-

tax state income annually but also aims to optimize the mmWave frequency with a broader bandwidth, which will enhance Indonesia's value. The application of spectrum usage fees with the new formula in industrial areas will have a multiplier effect on the output of the national economy and supporting Industry 4.0 and encourage digital economic growth and national digital transformation, especially for vertical industries in Indonesia.

## 8. Conclusions

In this paper, we have proposed a new formula for calculating the spectrum usage fee for 5G-mmWave spectrum implementation for the private network industry in Indonesia. The main basis for developing the proposed new formula is the adoption of the INDI 4.0 score for five priority industrial sectors in Indonesia. Along with this factor, we also considered the number of workers in the industry and combined them with the ITU-R SM.2012-5 (06/2016) framework.

We tested the proposal by applying the new formula to the Jakarta industrial area, assuming that the 5G-mmWave private network would be deployed in two industrial areas: Pulogadung and KBN.

The results show that the new formula always gives a lower spectrum usage fee than the current formula, which is advantageous to 5G-mmWave private network service operators. Such a saving can be regarded as a government subsidy for service operators to apply to various use cases in the industry, providing further economic benefits. Another advantage is that the new formula reflects a combination of the spirit of industrialization and the spirit of digital transformation.

Using an I–O model, we prove that while the proposed new formula produces a lower spectrum usage fee value, which means a loss of state income, it will generate a much greater positive effect on the output of the national economy.

*Future Studies*

Further studies need to focus on refining the costs of spectrum usage fees and 5G-mmWave private network infrastructure deployment for all industrial areas in Indonesia, not only Jakarta.

**Author Contributions:** Conceptualization, A.H., K.R. and M.S.; methodology, A.H., K.R., M.S. and I.K.R.; software, A.H.; validation, K.R., I.K.R., M.S. and M.Z.; formal analysis, A.H. and M.S.; investigation, A.H. and M.S.; resources, A.H.; data curation, A.H. and M.S.; writing—original draft preparation, A.H.; writing—review and editing, K.R. and M.S.; visualization, A.H. and M.S.; supervision, K.R. and M.S.; project administration, A.H.; funding acquisition, K.R. and A.A.P.R. All authors have read and agreed to the published version of the manuscript.

**Funding:** This publication is supported by Research Grant of *Hibah PUTI Q2* Universitas Indonesia, 2022 under contract number NKB-677/UN2.RST/HKP.05.00/2022.

**Institutional Review Board Statement:** Not applicable.

**Informed Consent Statement:** Not applicable.

**Data Availability Statement:** Not applicable.

**Acknowledgments:** The authors are grateful to the editors and reviewers for their invaluable contributions.

**Conflicts of Interest:** The authors declare no conflict of interest.

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
