# Peer review of "A Proposal for Formulating a Spectrum Usage Fee for 5G Private Networks in Indonesian Industrial Areas"

_informatics, doi:10.3390/informatics9020044_

Round 1

Reviewer 1 Report

This paper proposes a framework leading to the formulation of a new spectrum usage fee for 5G-mmWave implementation in the industrial areas of Indonesia. The paper presents an overview of demand response services. The paper is not organized as a research paper and does not show research results, it presents only a very simple formula without presenting a deep technical study.

The paper presents some well-known and basic notions and lacks novelty. The authors should remove all the well-known details and give more attention to explain the new contribution.

The proposed study is very basic and widespread, and the paper did not show any technical achievement compared to the state of the art.

The theoretical part of the paper should be more developed.

The paper does not determine the problematic explicitly and its resolution clearly.

The presentation quality of the paper should be improved for example: the format of the numbers in the whole paper needs to be corrected.

Line 378, the number of figure should be corrected.

The quality of the most figures in the paper is very poor and needs to be improved.

The proposed study should be extended and compared to other more recent studies in the literature.

The evaluation of the proposed framework should be given and discussed, and the validation test of the proposed approach is very important to show the performance in a real case.

The English of the paper and redaction style should be improved. Several orthographic and grammatical typos must be corrected.

Author Response

Response to Reviewer 1 Comments

Original Manuscript ID     : informatics-1668496

Original Article Title          : “A Proposal for Formulating a Spectrum Usage Fee for 5G Private Networks in Indonesian Industrial Areas”.

Part 1:

This paper proposes a framework leading to the formulation of a new spectrum usage fee for 5G-mmWave implementation in the industrial areas of Indonesia. The paper presents an overview of demand response services. The paper is not organized as a research paper and does not show research results, it presents only a very simple formula without presenting a deep technical study.

Response 1a:

Thank you for the response that has been given.

The paper is organized as a research paper (lines 121-126, page 3).

The paper is organised as follows: Section II explains the underlying theories. Section III focuses on the Indonesian profile, including its regulatory industrial ecosystem. Section IV evaluates the current formula of spectrum usage fees in Indonesia, while section V Methods and Proposed New Formula. Section VI focuses on the Results. Section VII discussion, and finally, section VIII concludes the study and suggests future studies.

Response 1b:

The results or contributions from this paper are (lines 110-120, page 3):

This paper makes the following key contributions:

  • It proposes the formulation of a new spectrum usage fee for 5G-mmWave private network implementation in Indonesian industrial areas.
  • The framework can be used as a recommendation for Indonesian regulatory policymakers as the country will start deploying 5G-mmWave private networks in the near future. The proposed formula reflects a comprehensive policy for supporting industrialisation and digital transformation and enables us to estimate the economic multiplier for 5G-mmWave private networks deployment in industrial areas.
  • The new spectrum usage fee approach provides an easy and direct way to price spectrum, so it can be used as a benchmark for other countries to apply spectrum usage fees to 5G-mmWave private networks in industrial area.

Response 1c:

It presents only a very simple formula without presenting a deep technical study.

Thank you for the response that has been given.

The authors add some paragraphs to explain a deep technical study. Figure 8 shows the new spectrum usage fee model for this study to present a deep technical analysis.

Figure 8. Indonesia spectrum usage fee Model (line 412, page 12).

  • Authors developed the proposed formula mainly by adopting the INDI 4.0 scores for Indonesia’s five priority industrial sectors (The score was obtained by surveying 303 companies in the five priority industrial sectors [46].) (Lines 423-425, page 12).
  • After considering such factors, we also took into account the number of workers in the industry and combined them with the ITU-R SM.2012-5 (06/2016) framework. (Lines 96-98, page 2), because the number of workers might not be proportional among different industries in the five priority sectors (Lines 446-447, page 13).
  • From the perspective of the engineering model, the primary focus of deployment in the 5G rollout is network densification, especially as mmWave bands with worse propagation properties (Lines 204-206, page 5). From an economic perspective, the cost of mmWave band spectrum densification in 5G-mmWave private network infrastructure is heavily influenced by the required CAPEX and OPEX [22], [29], [30]. (Lines 235-237, page 6).
  • To prove our argument, we develop the I-O model to show how 5G-mmWave private network implementation (Lines 526-527, page 14). The I-O model is often used in the analysis of industrial systems or macroeconomic systems to examine the structure of intersectoral linkages (Lines 532-533, page 15).

Point 2:

The paper presents some well-known and basic notions and lacks novelty. The authors should remove all the well-known details and give more attention to explain the new contribution.

Response 2:

Thank you for the response that has been given.

This paper being unique with the case in Indonesia, this paper contributes to being used as a reference for any service operator who launches 5G-mmWave private network in Industrial Area during the 5G era and also implies for Indonesian regulatory policymakers as the country will start deploying 5G-mmWave private networks in the near future.

This paper has three contributions (Lines 110-121, page 3).

  • It proposes the formulation of a new spectrum usage fee for 5G-mmWave private network implementation in Indonesian industrial areas.
  • The framework can be used as a recommendation for Indonesian regulatory policymakers as the country will start deploying 5G-mmWave private networks in the near future. The proposed formula reflects a comprehensive policy for supporting industrialisation and digital transformation and enables us to estimate the economic multiplier for 5G-mmWave private networks deployment in industrial areas.
  • The new spectrum usage fee approach provides an easy and direct way to price spectrum, so it can be used as a benchmark for other countries to apply spectrum usage fees to 5G-mmWave private networks in industrial area.

Point 3:

The proposed study is very basic and widespread, and the paper did not show any technical achievement compared to the state of the art.

Response 3:

Thank you for the response that has been given.

The technical achievement resulting from this paper is to create a new formula, which is shown in equation 6 (line 455, page 13). This paper explains how to get a new formula by modeling the current formula with the parameters from ITU-SM 2012, the value of the INDI 4.0 Score, and the private network parameter with limited geographical conditions. These steps can be seen in Figure 8.

The result of this new formula is that a Calt factor will affect the nominal IDR to be paid by the service operator to the government.

The result shows that the new formula always gives a lower spectrum usage fee than the current formula, which benefits 5G-mmWave private network service operators. Such savings can be regarded as a government subsidy for the service operators to use in various ways in the industry, providing further economic benefits. (Lines 650-653, page 19).

To prove our argument, the authors calculate the CAPEX and OPEX costs for 5G-mmWave private network deployments located in two Industrial areas (Pulogadung, KBN) using the Engineering-economic Model based on the new spectrum usage fee formula from the previous output after that we develop the I-O model to show how 5G-mmWave private network implementation in Pulogadung and KBN industrial areas will produce an economic impact greater than the state income lost. The dimension of the Input-Output (I-O) matrix used is 17 x 17 (Lines 406-409, page 12). Therefore, this simulation estimates the contribution of 5G-mmWave private networks implementation to GDP.

Point 4:

The theoretical part of the paper should be more developed.

Response 4:

Thank you for the response that has been given.

The author adds four paragraphs and three references to explain theoretical part about the solution regarding mmWave's weaknesses in 5G-mmWave private network engineering Economic Model section (Lines 218-234, pages 6):

There may be coverage problems in Pulogadung and KBN industrial areas covered by mmWave gNB during the initial deployment phase of the 5G-mmWave private network. Due to the weakness of reflected and diffracted signals, the propagation will rely heavily on LoS coverage. Indoor users expect to be served by mmWave gNB in the 5G-mmWave private network, which will provide exceptionally high capacity and the necessary resources for higher data rates. Increased penetration loss at high frequencies is one of the challenges in this context [26]. 

Because of penetration loss and atmospheric loss, a highly directional antenna is recommended for a 5G-mmWave private network to compensate for additional losses due to path loss in the industrial area[26].

Beamforming will ensure reliable links for deploying 5G-mmWave private networks when combined with high gain/large array antennas at the transmitter and receiver ends [27].

The ability of 5G-mmWave private networks to handle a large number of IoT devices in the industrial area is a crucial design goal. The development of extreme densification of cellular networks and additional spectrums are two possible solutions to meet these demands [28].

The authors add 3 new references below:

[26]        N. Al-Falahy and O. Y. K. Alani, “Millimetre wave frequency band as a candidate spectrum for 5G network architecture: A survey,” Phys. Commun., vol. 32, no. 2019, pp. 120–144, 2019.

[27]        G. E. Athanasiadou, P. Fytampanis, D. A. Zarbouti, G. V. Tsoulos, P. K. Gkonis, and D. I. Kaklamani, “Radio network planning towards 5g mmwave standalone small-cell architectures,” Electron., vol. 9, no. 2, pp. 1–10, 2020.

[28]        S. Tripathi, N. V. Sabu, A. K. Gupta, and H. S. Dhillon, “Millimeter-Wave and Terahertz Spectrum for 6G Wireless,” pp. 83–121, 2021.

Point 5:

The paper does not determine the problematic explicitly and its resolution clearly.

Response 5:

Thank you for the response that has been given.

The Abstract has been added several sentences explaining the problems encountered and how to solve them (Lines 17-23, page 1).

As spectrum usage fees are proportional to the width of the bandwidth, the current formula would result in an extremely high price when applied to 5G-mmWave private network, having a direct consequence of cost burden on the service operator. In this paper, we propose the formulation of a new spectrum usage fee for 5G-mmWave private network implementation in Indonesian industrial areas. To do so, we evaluate the current formula, adopt the framework offered by the ITU-R SM.2012-5 (06/2016), and use the industrial reference index – the Indonesia Industry Readiness Index 4.0 (INDI 4.0) score

Point 6:

The presentation quality of the paper should be improved for example: the format of the numbers in the whole paper needs to be corrected.

Response 6:

Thank you for the valuable suggestion.

The author has fixed the following sections:

  1. Equations format and number sequences of equations. (Lines 152, 338, 343, 351, 359, 455, 456).
  2. Tables number order and table format. (Lines 162, 313, 360, 368, 437, 540, 603).
  3. Figures number order. (Lines 145, 174, 261, 274, 290, 334, 380, 412).

Point 7:

Line 378, the number of figure should be corrected.

Response 7:

Thank you for the valuable suggestion.

Line 378 is the number for the figure title "Figure 7. The trend of the operators' profit as a result of 5G implementation in Indonesia [36]". This number has been corrected according to the order of the figures, as described in the order below:

Figure 1. 5G Usage Scenario [11] (line 145)

Figure 2. 5G private network [14] (line 174)

Figure 3. Contribution of internet users by region from all users [34] (line 261)

Figure 4. The proportion of each service operator's bandwidth in Indonesia [36]. (Line 274)

Figure 5. Spectrum usage fees of the three biggest service operators in Indonesia (IDR Million) [39]. (Line 290)

Figure 6. Spectrum valuation formula from ITU-R SM.2012-5 (06/2016) adapted from the current Indonesia spectrum usage fee. (Line 334)

Figure 7. The trend of the operators' profit as a result of 5G implementation in Indonesia [39]. (Line 380)

Figure 8. Indonesia spectrum usage fee model. (Line 412)

Point 8:

The quality of the most figures in the paper is very poor and needs to be improved.

Response 8:

Thank you for the valuable suggestion.

The author has corrected the following images:

  1. Figures 1 and 8, have been replaced and enlarged in size. (Lines 145 and 412).
  2. Figure 2 - Figure 7 has been enlarged in size (Lines 174, 261, 274, 290, 334, 380).

Point 9:

The proposed study should be extended and compared to other more recent studies in the literature.

Response 9:

Thank you for the valuable suggestion.

The Introduction has been added several sentences explaining about proposed study and compared to other more recent studies in the literature (Lines 62-68, page 2).

“Previous studies of spectrum usage fees may be found in [4], [5], [6], which look at how such fees are calculated for mobile cellular service (2G/3G) in Taiwan and Indonesia. These countries’ formulas differ based on various factors, including rivalry among service providers, geographic location, population, bandwidth, and expected government income from spectrum usage fees, among others. However, these studies have not discussed a spectrum usage fee that uses technology to access an mmWave frequency band with extensive bandwidth, and a limited distance.”

The authors add two references from the Previous Study (references no [9] and [10] in the Introduction (Lines 69-71, page 2).

“In other studies [7], [8], [9], [10] the authors only compare different pricing methods for private LTE and 5G networks using Finland as an example country but with narrowband bandwidth (10 MHz) and mid-band frequency (3.5 GHz).”

References:

[9]          M. Matinmikko-Blue, S. Yrjölä, V. Seppänen, P. Ahokangas, H. Hämmäinen, and M. Latva-Aho, “Analysis of Spectrum Valuation Approaches: The Viewpoint of Local 5G Networks in Shared Spectrum Bands,” 2018 IEEE Int. Symp. Dyn. Spectr. Access Networks, DySPAN 2018, 2019.

[10]        M. Matinmikko-Blue, S. Yrjola, V. Seppanen, P. Ahokangas, H. Hammainen, and M. Latva-aho, “Analysis of Spectrum Valuation Elements for Local 5G Networks: Case Study of 3.5 GHz Band,” IEEE Trans. Cogn. Commun. Netw., vol. 11000, no. Otakaari 24, pp. 1–1, 2019.

Point 10:

The evaluation of the proposed framework should be given and discussed, and the validation test of the proposed approach is very important to show the performance in a real case.

Response 10:

Thank you for the valuable suggestion.

The Abstract has been added several sentences explaining the evaluation of the proposed framework and validation test of the proposed approach.

Using the input-output model, we prove that despite the proposed formula bringing a lower spectrum usage fee, which means a loss in state income, it will lead to a much greater positive economic impact on the national GDP. Applying the new formula will eventually have a multiplier effect on various sectors and encourage digital economic growth and national digital transformation, especially for vertical industries in Indonesia. This study contribution can serve as a guideline or initial reference for Indonesian policymakers and service operators on applying the CAPEX and OPEX cost of using the new spectrum for 5G-mmWave private network service implementation and estimating the economic multiplier for 5G-mmWave private network service deployment in industrial areas (Lines 28-38, page 1). It can also be used as a benchmark case for other countries to apply spectrum usage fees for private networks in industrial area.

Point 11:

The English of the paper and redaction style should be improved. Several orthographic and grammatical typos must be corrected.

Response 11:

Thank you for the valuable suggestion.

Grammatical mistakes and typographical errors already fixed through the proofreading process.

Reviewer 2 Report

The paper presents proposed new formula for finding the fee of spectrum usage in Indonesia. Several criteria are taken into account and the formula is used for calculation the fee in two industrial sectors. The findings point out that this approach leads to lower fee. Benefits and  advantages are also discussed.

  • I am confused with Tables numeration and captions- for example see Tables 5 and 6. 
  • Also, the numeration of Figures should be checked - Figure 2 is mentioned twice.  Figure 3 should be before Figure 4.
  • I recommend the text style in all Tables to be equal. Figure 1 should be larger for better reading.
  • The formulas style should be the same - please give some space before and after all equations.
  • Section 5 is too short - maybe it could be added to the section 6.
  • The future work should be discussed in brief in the last section of the paper.

Author Response

Response to Reviewer 2 Comments

Original Manuscript ID     : informatics-1668496

Original Article Title          : “A Proposal for Formulating a Spectrum Usage Fee for 5G Private Networks in Indonesian Industrial Areas”.

Point 1:

I am confused with Tables numeration and captions- for example see Tables 5 and 6

Response 1:

All table numbering and table captions have been corrected. The table order and table captions are as follows:

After corrected

Table 1. Peak data rate vs Modulation for 26/28 GHz. (Line 162).

Table 2. The number of operational industrial areas in 2019 [38]. (Line 313).

Table 3. The current formula for Indonesian spectrum usage fee [2]. (Line 360).

Table 4. The Indonesian basic price index value (I) for radio frequency bands [2]. (Line 368).

Table 5. CINDI score for the five priority industrial sectors scores [42]. (Line 437).

Table 6. Spectrum usage fee comparison between the current and new formula in five priority industries (IDR) and 5G-mmWave private network deployment costs for the 10-year duration of the spectrum usage fee. (Line 540).

Table 7. Multiplier and economy impact from 5G-mmWave deployment investment (Million IDR). (Line 603).

Before corrected

Table 1. Peak data rate vs. Modulation for 26/28 GHz.

Table 2. The Number of operational industrial areas in 2019 [38].

Table 3. The Current formula for Indonesian spectrum usage fee [2].

Table 4. The Indonesian basic price index value (I) for radio frequency bands [2].

Table 2 CINDI score for the five priority industrial sectors scores [42].

Table 3 Spectrum usage fee comparison between the current and new formula in five priority industries (IDR) and 5G deployment costs for the 10-year duration of the spectrum usage fee.

Table 4. Multiplier and economy impact from 5G deployment investment (Million IDR)

Point 2:

Also, the numeration of Figures should be checked - Figure 2 is mentioned twice.  Figure 3 should be before Figure 4.

Response 2:

Thank you for the valuable suggestion.

The figures' numbering has been corrected. Figure 3 (line 261) has been placed before Figure 4 (line 274). Figure 2 has been mentioned once (line 171).

Point 3:

I recommend the text style in all Tables to be equal. Figure 1 should be larger for better reading.

Response 3:

Thank you for the valuable suggestion.

All table text styles have been corrected and adapted to the MDPI Informatics format. In addition (Lines 162, 313, 360, 368, 437, 540, 603), figure 1 has been modified and enlarged (Line 162).

Point 4:

The formulas style should be the same - please give some space before and after all equations.

Response 4:

Thank you for the valuable suggestion.

The style of the formula has been changed using the equation editor. In addition, more space has been given before and after the equations.

Equation 1 (line 152)

Equation 2 (line 338)

Equation 3 (line 343)

Equation 4 (line 351)

Equation 5 (line 359)

Equation 6 (line 455)

Equation 7 (line 456)

Point 5:

Section 5 is too short - maybe it could be added to the section 6.

Response 5:

Thank you for the valuable suggestion.

The title was changed to "Methods and Proposed New Formula" after being merged into Chapter 5. Methods and Proposed New Formula (Line 392, page 11),

Point 6

The future work should be discussed in brief in the last section of the paper.

Response 6:

Thank you for the valuable suggestion.

The authors have added further studies in the last section of the paper with the following sentence:

“Further studies need to focus on refining the costs of spectrum usage fees and 5G-mmWave private network infrastructure deployment for all industrial areas in Indonesia, not only Jakarta. “ (Line 660-662, Page 20).

Reviewer 3 Report

The topic of this communication entitled: "A Proposal for Formulating a Spectrum Usage Fee for 5G Private Networks in Indonesian Industrial Areas" falls within the profile and scope of the Informatics.

The article discusses spectrum usage fees for 5G-mmWave implementation in Indonesian industrial areas.

Recommendation – Consider after minor changes

Comment:

Authors in the manuscript pointed up that uncritical adopting of existing formulas to 5G-mmWave spectrum fee would result in an illegitimate high price. To argue this the limited coverage distances were taken into account to formulate the proposed spectrum fee. However, with mmWave frequency range there exists another important disadvantage.  The mm-waves cannot penetrate through the building walls due to very high energy absorption. From this reason it is rather impossible to realize communications inside the buildings so the communications as Internet of Thing (or smart factories) is questioned. This aspect should be explained when discussing about the budget.

Author Response

Response to Reviewer 3 Comments

Original Manuscript ID     : informatics-1668496

Original Article Title          : “A Proposal for Formulating a Spectrum Usage Fee for 5G Private Networks in Indonesian Industrial Areas”.

Point 1:

With mmWave frequency range there exists another important disadvantage.  The mm-waves cannot penetrate through the building walls due to very high energy absorption. From this reason it is rather impossible to realize communications inside the buildings so the communications as Internet of Thing (or smart factories) is questioned. This aspect should be explained when discussing about the budget.

Response 1:

Thank you for the valuable suggestion.

The author adds four paragraphs and three references to explain the solution regarding mmWave's weaknesses in 5G-mmWave private network Engineering Economic Model section (Lines 218-234, pages 6):

There may be coverage problems in Pulogadung and KBN industrial areas covered by mmWave gNB during the initial deployment phase of the 5G-mmWave private network. Due to the weakness of reflected and diffracted signals, the propagation will rely heavily on LoS coverage. Indoor users expect to be served by mmWave gNB in the 5G-mmWave private network, which will provide exceptionally high capacity and the necessary resources for higher data rates. Increased penetration loss at high frequencies is one of the challenges in this context [26].  

Because of penetration loss and atmospheric loss, a highly directional antenna is recommended for a 5G-mmWave private network to compensate for additional losses due to path loss in the industrial area[26].

Beamforming will ensure reliable links for deploying 5G-mmWave private networks when combined with high gain/large array antennas at the transmitter and receiver ends [27].

The ability of 5G-mmWave private networks to handle a large number of IoT devices in the industrial area is a crucial design goal. The development of extreme densification of cellular networks and additional spectrums are two possible solutions to meet these demands [28].

The authors add 3 new references below:

[26]        N. Al-Falahy and O. Y. K. Alani, “Millimetre wave frequency band as a candidate spectrum for 5G network architecture: A survey,” Phys. Commun., vol. 32, no. 2019, pp. 120–144, 2019.

[27]        G. E. Athanasiadou, P. Fytampanis, D. A. Zarbouti, G. V. Tsoulos, P. K. Gkonis, and D. I. Kaklamani, “Radio network planning towards 5g mmwave standalone small-cell architectures,” Electron., vol. 9, no. 2, pp. 1–10, 2020.

[28]        S. Tripathi, N. V. Sabu, A. K. Gupta, and H. S. Dhillon, “Millimeter-Wave and Terahertz Spectrum for 6G Wireless,” pp. 83–121, 2021.

Round 2

Reviewer 1 Report

The authors have satisfactorily addressed my concerns.

The authors have done several efforts to improve the technical quality of the paper. For this reason, a minor revision is needed:

-The authors should provide a separate paragraph to discuss the related works. This paragraph should not be included in the introduction.

-The quality of figures should be improved.

-The format of numbers in the paper should be corrected.

-The evaluation of the proposed system should be done by the comparisons of the obtained results to the recent state of the art. Indeed, the comparisons should be based on the obtained results not only on the differences between the proposed system and related work.

-The authors should organize the whole paper as a research paper which respects the publication standard.

-The English of the paper and redaction style should be improved.

Author Response

Response to Reviewer 1 Comments- Round 2

Original Manuscript ID     : informatics-1668496

Original Article Title          : “A Proposal for Formulating a Spectrum Usage Fee for 5G Private Networks in Indonesian Industrial Areas”.

Point 1:

The authors should provide a separate paragraph to discuss the related works. This paragraph should not be included in the introduction.

Response 2:

Thank you for the valuable suggestion.

The authors provide paragraphs in section 2 Underlying Theories (Pages 5, Lines 186-199)

5G Spectrum Usage Fee

Previous studies of spectrum usage fee include [4]–[6], which examine the formulation of spectrum usage fees for service operators (2G/3G) in Taiwan and Indonesia. The formulas used by these nations vary depending on various factors, including competition among service operators, geographical condition, population, bandwidth, and expected income from spectrum usage fees for the government. The goals of each nation are the same – namely to formulate a fair and technology-neutral fee model to enhance the efficiency of spectrum utilization. The framework was developed based on frequency bands below 3 GHz, operated on narrow bandwidths up to 20 MHz for the 4G network.

 5G provides three different usage scenarios that require simultaneous access to low, mid, and high bands to meet 5G capacity, latency, coverage, and quality requirements to comply with advanced spectrum range use cases [20], [21]. Therefore, the formulation of the spectrum usage fee may have a different focus for different bands. Recent research has begun to conduct spectrum valuation and pricing of private LTE and 5G networks at the 3.5 GHz frequency band  [7]–[10]..

Point 2:

The quality of figures should be improved.

Response 2:

Thank you for the valuable suggestion.

The author has fixed the following figures:

Figure 1 has been changed and enlarged (Line 142).

Figure 2 has been enlarged in size. (Line 171).

Figure 3 has been enlarged in size. (Line 259).

Figure 4 has been changed and enlarged. (Line 274).

Figure 5 has been changed and enlarged. (Line 290).

Figure 6 has been enlarged in size. (Line 336).

Figure 7 has been changed and enlarged. (Line 384).

Point 3:

The format of numbers in the paper should be corrected.

Response 3:

Thank you for the valuable suggestion.

The format of numbers in the paper has been fixed according to the MDPI format below:

Figure 1. MDPI Format number

This is the paper’s format number according to the MDPI format:

  1. Introduction
  2. Underlying Theories

2.1.      5G-mmWave Technology  

2.2.      5G Private Network for Industrial Areas

2.3.      5G Spectrum Usage Fee     

2.4.      5G-mmWave private network Engineering Economic Model

  1. Indonesian Profile

3.1.      Mobile Ecosystem and Spectrum Regulatory Policy   

3.2.      The Indonesian Industry   

  1. Evaluation of Current Formula for Spectrum Usage Fee

4.1.      Origin of the Current Formula     

4.2.      Problem Of The Current Formula Used For 5G-mmWave      

  1. Methods and Proposed New Formula

5.1.      Methods       

5.2.      Proposed New Formula for Spectrum Usage Fee         

5.2.1.   Significance of Industrial Reference Index         

5.2.2.   Considering the Number Of Workers In The Industry

5.2.3.   New Formula          

  1. Results
  2. Discussion

7.1.      Input-Output (I-O) Model 

7.2.      Economic Impact Of 5G-mmWave Private Network In Industrial Areas      

7.3.      Advantages Of The New Formula           

  1. Conclusion and Future Studies

Point 4:

The evaluation of the proposed system should be done by the comparisons of the obtained results to the recent state of the art. Indeed, the comparisons should be based on the obtained results not only on the differences between the proposed system and related work.

Response 4:

Thank you for the valuable suggestion.

The result section (Section 6) demonstrates proof of concept by comparing the proposed formula results to the recent state-of-the-art for calculating the spectrum usage fee in Indonesian industrial areas (Page 14, Lines 469-471)

The explanation below demonstrates how to obtain such spectrum usage fee values. The calculation example is for the chemistry sector in the industrial area of Pulogadung.

To acquire the value of the spectrum usage fee using the current formula (the recent state of the art) (Table 6 – row (12) column (3))

N*K                                        = 28.36

Index Price (I)                       = IDR 2,383,-

Kilopopulation (C)             = 65

Bandwidth (B)                      = 100 MHz

Therefore, the current spectrum usage fee value is

= N*K*I*C*B

= 28.36 * IDR 2,383 * 65 * 100

= IDR 439,221,848 per year or IDR 4,392,218,48 for 10 years

(Rounded to IDR 4,392 million for the 10-year fee)

To acquire the value of the spectrum usage fee using the new formula (proposed formula) for the chemistry industry (Table 6 – row (6) column (3))

N*K                                        = 28.36

Index Price (I)                       = 2,383,-

Kilopopulation (C)             = 65

Bandwidth (B)                      = 100 MHz

CAlt                                        = 0.16844

Therefore, the current spectrum usage fee value for chemistry in Pulogadung is

= N*K*I*C*CAlt chemistry*B

= 28.36 *IDR 2,383 * 65 *0.16844* 100

= IDR 73,983,868 per year or IDR 739,838,681 for 10 years

(Rounded to 740 million for the 10-year fee)

Point 5:

The authors should organize the whole paper as a research paper which respects the publication standard.

Response 5:

Thank you for the valuable suggestion.

According to the MDPI format the paper organize with this sections below:

  1. Introduction

The introduction should briefly place the study in a broad context and highlight why it is important. It should define the purpose of the work and its significance. The current state of the research field should be carefully reviewed and key publications cited.

  1. Materials and Methods

The Materials and Methods should be described with sufficient details to allow other’s to replicate and build on the published results.

  1. Results

This section may be divided by subheadings. It should provide a concise and precise description of the experimental results, their interpretation, as well as the experimental conclusions that can be drawn.

  1. Discussion

Authors should discuss the results and how they can be interpreted from the per-spective of previous studies and of the working hypotheses.

  1. Conclusions

This section is not mandatory but can be added to the manuscript if the discussion is unusually long or complex.

Authors already organize the whole paper research paper as follow:

  1. (Lines 41-123)

This section explain about Spectrum issues, Previous studies of spectrum usage fees, The current formula does not work well for calculating 5G-mmWave private network spectrum usage fees, for this reason, Indonesia needs a new formula for spectrum usage fees, which would result in a more financially acceptable and affordable 5G-mmWave private network service operator      

  1. Underlying Theories. (Lines 124-242)

This section explain about theories including 5G-mmWave Technology, 5G Private Network for Industrial Areas, 5G Spectrum Usage Fee, 5G-mmWave private network Engineering Economic Model.          

  1. Indonesian Profile (Lines 243-324)

This section explain about Mobile Ecosystem and Spectrum Regulatory Policy and The Indonesian Industry ecosystem.

  1. Evaluation of Current Formula for Spectrum Usage Fee (Lines 324-395)

This section explain about Origin of the Current Formula and Problem of the Current Formula Used for 5G-mmWave.     

  1. Methods and Proposed New Formula (Lines 397-468)

This section explain about Methods, Proposed New Formula for Spectrum Usage Fee considering Significance of Industrial Reference Index and Considering the Number of Workers in the Industry, and how to get the New Formula

  1. Results. (Lines 469-519)

In this section, authors demonstrate proof of concept by testing the new formula for calculating the spectrum usage fee in Indonesian industrial areas.      

  1. Discussion. (Lines 520-641)

It is clearly indicate that the proposed formula will always produce a lower value than the current formula for the spectrum usage fee. Such a value would have a positive impact on the industry as operators would pay a cheaper price and make the economic cost more efficient.

This section explain about, developing the I-O model to show how 5G-mmWave private network implementation, Economic Impact Of 5G-mmWave Private Network in Industrial Areas and Advantages Of The New Formula           

  1. Conclusion and Future Studies (Lines 642-663)

Point 6:

The English of the paper and redaction style should be improved.

Response 6a:

Thank you for the valuable suggestion.

We have submitted this paper to the native proofreading service before, but to improve it, we will send it to the Language Editing Services from MDPI for proofreading.

Response 6b:

Thank you for the valuable suggestion. The redaction style has been improved in this section:

  • The authors have changed the table contents style from justifying to center in table 1-7 (Line 159, 315, 362, 372, 442, 543, and 604).
  • The subsection for all sections has changed from Normal style to italic style (Lines 125, 161, 185, 200, 244, 292, 325, 373, 398, 418, 535, 555, 607).
  • The citation style has been improved; for example, the previous style was as follows [4], [5], [6], so the style is now as follows [4]-[6]. The author uses the Mendeley desktop for reference management (Line 61, 68, 186, 199, and 290).
